# Exogenous Arachidonic Acid Affects Fucoxanthin Biosynthesis and Photoprotection in *Phaeodactylum tricornutum*

**DOI:** 10.3390/md20100644

**Published:** 2022-10-17

**Authors:** Shuaiqi Zhu, Song Bin, Wenda Wang, Shan Lu, Wenqiang Yang

**Affiliations:** 1State Key Laboratory of Pharmaceutical Biotechnology, School of Life Sciences, Nanjing University, Nanjing 210023, China; 2Photosynthesis Research Center, Key Laboratory of Photobiology, Institute of Botany, Chinese Academy of Sciences, Beijing 100093, China; 3University of Chinese Academy of Sciences, Beijing 100049, China; 4Innovative Academy of Seed Design, Chinese Academy of Sciences, Beijing 100093, China

**Keywords:** gene expression, carotenoid, principal component analysis, light harvesting, non-photochemical quenching

## Abstract

Fucoxanthin is an oxygenated carotenoid component that has been reported to play important roles in anti-oxidation, anti-obesity and anti-cancer in the human body. Fucoxanthin-chlorophyll protein (FCP) complexes participate in light harvesting and photoprotection in diatom. In order to better understand the change of fucoxanthin content and its role in photoprotection, the growth, fucoxanthin biosynthesis and photosynthetic phenotypes were studied in the diatom *Phaeodactylum tricornutum* under the treatment of exogenous arachidonic acid (AA). Our results demonstrated that even low concentration of AA at 0.1 mg/L strongly induced fucoxanthin accumulation in algal cells to a maximum of 1.1 mg/g dry weight (DW), which was 36.6% higher than that in the untreated ones. By principal component analysis (PCA), we also identified a close correlation between fucoxanthin accumulation and the expression of genes involved in fucoxanthin biosynthesis, especially phytoene synthase (PSY), suggesting that AA change the metabolism of fucoxanthin by inducing carotenoid metabolic enzymes at the transcriptional level. Furthermore, we found that the exogenous application of AA affected non-photochemical quenching (NPQ) and photoinhibition, which resulted from the changed diadinoxanthin (DD) and diatoxanthin (DT) cycle, and thus played an important role in photoprotection.

## 1. Introduction

Carotenoids are a major group of natural pigments in land plants, algae, cyanobacteria and some fungi [1]. In photosynthetic organisms, carotenoids play important roles in both light harvesting and photoprotection [2]. In the past two decades, carotenoid metabolism has been largely deciphered, and both the metabolic pathways and the regulation mechanisms have been extensively reviewed [3]. Generally, the biosynthesis of carotenoids occurs in plastids, starting from the methylerythritol 4-phosphate (MEP) pathway. Geranylgeranyl diphosphate (GGPP) synthase (GGPPS) is a key metabolic enzyme shared by the biosynthesis of several downstream products, such as carotenoids, chlorophylls (Chl)s and tocopherols [4]. Phytoene synthase (PSY) is the first committed enzyme that directs metabolic flux from GGPP into carotenoid biosynthesis by converting GGPP into phytoene, which is then catalyzed by phytoene desaturase (PDS) into lycopene [3]. Various cyclases, isomerases and hydroxylases are involved in carotenoid metabolic pathway as well. Most of the land plants share common carotenoid constituents, of which β-carotene and lutein are usually the major species [3]. It is also demonstrated that carotenoids in algae and land plants are all generated from an identical central part of the carotenoid biosynthesis pathway [5,6]. However, new carotenoid derivatives with different structural modifications are frequently reported in specific plant species, such as astaxanthin in *Adonis aestivalis* and *Haematococcus pluvialis*, and loroxanthin in *Chamydomonas reihardtiii* [5,7]. Studies on carotenoid distribution among different algal groups uncover a broad variation and suggest a divergence of carotenoid biosynthesis pathway during early plant evolution. For example, in some primitive species of Rhodophyta, no carotenoid component with an ε-ring (e.g., lutein) was found, indicating an absence of lycopene ε-cyclase (LCYE) [5,8]. Heterokontes, including diatoms and brown algae, do not have carotenoids with ε-rings either, and all their carotenoids are derived from β-carotene [5].

Fucoxanthin is an oxygenated carotenoid with a characteristic orange color, and specifically distributed in certain algal species of diatoms, red and brown algae [9]. In diatoms and brown algae, fucoxanthin, as well as Chl, combines with apoproteins to form fucoxanthin-chlorophyll binding proteins (FCP)s, which have functions similar to those of the light harvesting complex (LHC) in higher plants [10,11,12]. Because of its unique molecular structure (i.e., an allenic bond and 5,6-mono-epoxide), fucoxanthin possesses multiple pharmaceutical activities in the treatment of cancer, inflammation and obesity [13,14,15,16]. Deduced from its structure, fucoxanthin is believed to be synthesized from neoxanthin [17]. However, genes and enzymes for its biosynthesis have not been fully cloned and characterized yet. Lack of biochemical and genetic information hampers the application of metabolic engineering to heterologously produce fucoxanthin by fermentation. In addition, chemical synthesis of fucoxanthin has not been successfully achieved either.

High light induces excessive excitation of the photosynthetic pigments, and leads to overreduction of the electron transport chains and generation of reactive oxygen species (ROS), which impairs the components in the photosynthetic machinery [18]. The reaction center D1 protein of photosystem (PS) II is the primary target of photodamage [19]. However, rapid D1 repair cycle is developed in plants and algae, which is an effective mechanism to protect PSII from photoinactivation [20]. Various other protection mechanisms have also been developed in plants and algae during evolution, among which non-photochemical quenching (NPQ) is one of the most important photoprotective mechanisms [18]. NPQ is related to the xanthophyll cycle in higher plants and green algae [21], while in diatom, to the diadinoxanthin (DD) and diatoxanthin (DT) cycle [18]. Energy-dependent quenching (qE), the most important component of NPQ, reflects the increase of heat dissipation of excess light after activation of qE effector protein and change of pigment composition in PSII. In diatoms, LHCX, one of the high light response genes of the light-harvesting complex stress-related (LHCSR) family, is considered as qE effectors to participate in photoprotection, and there are 4 members in the LHCX family identified in the genome in *P. tricornutum* [22].

Chemical elicitors are widely used to promote the production of secondary metabolites in higher plants and algal cultures. For example, abscisic acid, salicylic and jasmonic acid were found to improve lipid accumulation in two freshwater *Chlorella* strains [23]. In addition, it has been shown that, in *Ginkgo biloba*, arachidonic acid (AA) promoted the accumulation of ginkgo lactone by regulating the expression of the gene encoding 1-deoxy-D-xylulose 5-phosphate synthase, a key enzyme in the MEP pathway, which produces both isopentenyl diphosphate and dimethylallyl diphosphate for carotenoid biosynthesis via GGPP [24].

*Phaeodactylum tricornutum* is a diatom to be widely investigated as a potential source for fucoxanthin production [25]. In addition, *P. tricornutum* can be cultivated conveniently because it grows rapidly. The genetic transformation system and the genome of *P. tricornutum* were published [26,27]. Therefore, *P. tricornutum* has been treated as a model species of diatom [28].

In diatom, fucoxanthin is the important carotenoid to light-harvesting, which is used to conform FCPs [2]. Previously, it was reported that fucoxanthin had an important role of photoprotection except for the function of light-harvesting because of the conformational change of FCP aggregation [18,29]. Two carotenoids, DD and DT, which constitute DD-DT cycle to protect the diatom from harmful ROS [30]. Furthermore, DT, as a quencher, can induce the detached and aggregated FCP complexes to influent NPQ [31]. Thus, DT has a crucial role in FCP aggregation. The question then arises whether and how fucoxanthin, which is the component of FCPs, affects DD-DT cycle, leading to the change of NPQ. This question was not reported previously. Based on this, we aim to investigate the relationship between the photosynthesis and photoprotection by adjusting production of fucoxanthin using exogenous chemical elicitor AA.

## 2. Results

### 2.1. Arachidonic Acid Modulates the Growth of P. tricornutum

In this study, we quantified the growth of *P. tricornutum* under different treatments by both measuring the absorbance at 680 nm and counting cell numbers under a microscope. There was a good correlation between these two approaches in the first 8 days of the culture without AA supplement. Our statistical analysis resulted in a linear equation of y = 0.0095x − 0.0005 (y represents the value of the OD_680_, x represents the number of cells) with a correlation coefficient R^2^ = 0.9962. After 4 days, the growth slowed down and a maximal cell concentration was reached at the 6th day. Treatment with different concentrations of AA showed different effects on the growth of *P. tricornutum* (Figure 1). Low concentrations of AA (0.1 to 62.5 mg/mL) helped maintaining the growth in the first 4 days, and further increased to the 6th day. However, the growth of *P. tricornutum* was significantly repressed by high concentration of AA (e.g., 312.5 mg/L), and started to decline after the 4th day (Figure 1).

### 2.2. Low Level of AA Favors Fucoxanthin Production

The results of HPLC analysis showed a fucoxanthin concentration of 0.805 mg/g dry weight (DW) in the 6th day in *f/2* medium without AA. Even a low concentration of AA (0.1 mg/L) induced the fucoxanthin production to 1.1 mg/g DW; while other concentrations of AA ranging from 2.5–62.5 mg/mL showed no effects on fucoxanthin production and a high concentration of AA (312.5 mg/L) strongly repressed the production (Table 1).

### 2.3. AA Affects the Expression of Carotenoid Metabolism Genes

By homology search, we successfully identified transcript sequences in *P. tricornutum* sharing high similarities with known genes for PSY, PDS, ζ-carotene desaturase (ZDS), carotenoid isomerase (CRTISO), lycopene β-cyclase (LCYB) and zeaxanthin epoxidase (ZEP) on carotenoid metabolic pathway (Table 2). Sequence homology analysis showed that most of these genes have the highest sequence similarities (above 65%) with corresponding homologs from the diatom *Fistulifera solari*, while the *PDS* gene of *P. tricornutum* shared the highest sequence similarity (60%) with *Chromochloris zofingiensis*—a member of green alga (Table 2). From the full transcriptome of *P. tricornutum*, we could not identify a transcript that share sequence similarity with known genes encoding LCYE or carotene ε-hydroxylase, indicated that *P. tricornutum* has a primitive carotenoid metabolic pathway without ε-ring derivative.

As the accumulation of fucoxanthin was induced by AA treatment, we quantified the expression of the genes identified in carotenoid metabolism pathway. Treatment with 0.1 mg/L AA promoted the expression of all these genes (*LCYB*, *PSY*, *PDS*, *CRTISO*, *ZDS* and *ZEP*) (Figure 2A). In all AA-treated samples, *ZEP* was expressed at higher levels compared with the control, suggesting that the algal cells encountered a stressed condition induced by the exogenous AA. PCA indicated major contribution of *PSY*, *PDS* and *CRTISO* genes to fucoxanthin production, compared with that of *LCYB*, *ZDS* and *ZEP* (Figure 2B). In addition, the contribution of *PSY* is the highest among these genes (Figure 2B).

### 2.4. Energy Transfer Is Affected by AA Treatment in P. tricornutum

In order to investigate the effect of fucoxanthin changes on the light absorbance and energy transfer, the absorption spectra in the region of 400–750 nm and the steady-state fluorescence spectra at 77 K were measured. There was no difference in the spectral shape of the algae cells with or without AA treatment, only a slight increase was present at the range of 513–660 nm under the treatment of AA, which may be originated from the increased content of fucoxanthin (Figure 3A), as a broad shoulder in the region of 513–580 nm may originate from fucoxanthin (Nagao et al., 2019).

The result of steady-state fluorescence spectra at 77 K showed that the intensity of the peak in 688 nm under the treatment of AA was higher than that of CK when normalized at 713 nm (Figure 3B), suggesting that energy transfer was not affected by AA treatment, while more energy was distributed to PSII, which might be resulted from the increase in fucoxanthin content.

### 2.5. The Activities of PSII and PSI Are Not Affected by AA Treatment

In order to investigate whether the activities of PSII and PSI were affected when the algae were treated with AA, Chl fluorescence and P700 redox kinetics were employed to detect. The value of *F_v_/F_m_* of the algae under the treatment of AA was similar to that of CK, while Y(II) was a little higher in AA-treated algae than that of CK when the light intensity of AL was lower than 300 μmol photons m^−2^ s^−1^ (Figure 4), suggesting that the electron transport through PSII was enhanced under the treatment of AA.

The quantum yield of PSI [Y(I)] in AA-treated cells was similar to that of CK (Figure 5A). In addition, the P700 redox state in AA-treated algae was not changed compared with that of CK when measured in the presence of 3-(3,4-dichloroprenyl)-1-1-dimethylurea (DCMU) and hydroxylamine (HA), which can block the electrons from PSII efficiently (Figure 5B). The above results indicated that the function of PSI and cyclic electric transfer were not affected under the treatment of AA.

### 2.6. Non-Photochemical Quenching Is Decreased under the Treatment of AA

In order to investigate whether non-photochemical quenching was changed in AA-treated *P. tricornutum*, which absorbed more excessive energy, a DUAL PAM-100 Chl fluorometer was employed to confirm variation in NPQ. The results showed that NPQ of AA-treated *P. tricornutum* was decreased significantly compared with that of CK (Figure 6A). NPQ in diatom is mainly linked to diadinoxanthin-diatoxanthin cycle, which is similar to xanthophyll cycle in higher plants and green algae [18,32,33,34]. The effect of DT as an antioxidant against the damage of reactive oxygen species under long-term high light is also proposed [2]. In order to understand the origin of the decreased NPQ, de-epoxidation states (DES), the ratio between DT and DT+DD, and the relative level of xanthophyll (DD+DT), which was normalized to the fucoxanthin content, were detected (Figure 6B). The value of DES and the relative level of xanthophyll (DD+DT) in AA-treated *P. tricornutum* were lower than that in CK (Figure 6B), which was in accordance with the decreased NPQ.

Light-harvesting complex stress-related proteins LHCXs are components of the peripheral fucoxanthin-chlorophyll protein complex [18]. LHCXs can be regulated under different light intensities and nutrient deficiency to modulate the capacity of NPQ [22]. In diatom genome, four LHCX family members were found, and the functional diversification of LHCX family participates in the regulation of multiple abiotic stress [22]. The relative expression levels of the LHCX family members were identified, and the results showed that the expression levels of all the four genes (*LHCX1*-*LHCX4*) were increased significantly in AA-treated cells, especially the expression of *LHCX1*, which was three times of that in the CK (Figure 6C). However, the protein level of LHCX1 in AA-treated cells was lower than that in CK, which was in accordance with the decrease in NPQ (Figure 6D).

### 2.7. Arachidonic Acid Treatment Induces Photoinhibition and Slows down Recovery of PSII Activity from Photoinhibition

To investigate whether AA treatment leads to photoinhibition, Y(NO), the quantum yield of non-regulated non-photochemical energy dissipation, was measured, and the result showed that the value of Y(NO) in AA-treated algae was higher than that in CK (Figure 7A), which indicated that AA-treated algal cells might suffer from photoinhibition. In the meantime, changes of PSII activity (*F_v_*/*F_m_*) during high light treatment (1000 μmol photons m^−2^ s^−1^) and subsequent dark recovery were also measured to monitor the photoinhibition process and the ability of the cells to recover from photoinhibition. The value of *F_v_/F_m_* declined in both AA-treated cells and the CK ones during high light exposure, and recovered during the following darkness adaptation (Figure 7B). PSII activity in AA-treated cells decreased faster and to a lower level when exposed to high light compared with the CK ones, and was still lower than the CK ones even after 1 h recovery in the dark (Figure 7B). In a word, the AA-treated algae are prone to suffer from photoinhibition.

## 3. Discussion

Biomass is one of the most important issues to be considered in the process of developing natural products. Previous studies have reported that the growth of *P. tricornutum* is affected by pH, temperature, carbon source and other factors [17,25]. In our work, we measured the growth of *P. tricornutum* under different AA concentrations, and found that whereas a high concentration of AA shortened the time of logarithmic growth and lowered the biomass, AA at a low concentration of 0.1 mg/L was sufficient to induce the growth of *P. tricornutum*. After 6 days, the biomass of 0.1 mg/L AA treated algae was 1.39-fold of the control.

Elicitors are exogenous chemicals that can increase the accumulation of metabolites in plants. Although they are usually not directly involved in the metabolic process, they are generally found to trigger the defensive responses in higher plants, which in turn, initiate the biosynthesis of defensive natural products (phytoalexins). In this study, HPLC analysis showed that AA at a concentration of 0.1 mg/L was sufficient to induce an increase in the production of fucoxanthin (Table 1). Gene expression analysis demonstrated that the induction of fucoxanthin biosynthesis by AA was at the transcriptional level, as significant increases in transcript abundance were found for genes involved in the central part of the carotenoid metabolism pathway (Figure 2A). Although it is not known how fucoxanthin is derived from neoxanthin, induction of a group of genes encoding enzymes infers an enhanced metabolite flux towards fucoxanthin biosynthesis. From PCA, we identified the major contribution of *PDS*, *PSY* and *CRTISO* to fucoxanthin production (Figure 2B). Among these enzymes, PSY is the enzyme that specifically directs metabolic flux from GGPP into carotenoid biosynthesis [4]. Previously, it was reported that the content of fucoxanthin increased by enhancing blue light and the addition of tryptone, the expression level of *PSY* gene was upregulated significantly [35]. The highest contribution of *PSY* to fucoxanthin accumulation indicates the transcriptional regulation of carotenoid metabolic pathway for fucoxanthin biosynthesis.

From our analysis, AA at higher concentrations did not show significant effects on fucoxanthin accumulation (Table 1). It is suggested that AA at high concentrations might result in a stress condition, which possibly inhibited corresponding metabolic processes. Transcripts of *ZEP* are generally found to accumulate when photosynthetic organisms are under high light stress [36]. The induction of *ZEP* expression by various concentrations of AA treatment further supported our postulation that AA treatment resulted in a stress condition, instead of promoting cell growth (Figure 1 and Figure 2). Given the fact that fucoxanthin is a component of FCPs in *P. tricornutum*, it is possible that the biosynthesis of fucoxanthin is modulated at transcriptional level for an adaptation to the stress raised by exogenous AA.

Compared with *Chlamydomonas reinhardtii,* that has multiple genes for lycopene β- and ε- cyclases, *P. tricornutum* has a relatively simple carotenoid metabolic pathway. The only gene for LCYB agrees with the absence of carotenoid species with ε-ring. LCYB is therefore not a branch point enzyme that modulates the metabolic flux between the β- and ε- branches. Thus, the result of qPCR and PCA analysis suggested that *LCYB* gene expression was not correlated with fucoxanthin production.

The pathway of carotenoid biosynthesis in *P. tricornutum* shows that neoxanthin is the branch point of fucoxanthin and DD [2,17]. In diatom, fucoxanthin is treated as light harvesting pigment, which is crucial to the efficiency of light capture, while DD-DT cycle is related to NPQ. However, whether the change of fucoxanthin content affects DD, and thus NPQ, needs to be discussed. The absorption spectra of the algae in 513–660 nm region was increased when treated with AA (Figure 3A), which may be originated from the increased content of fucoxanthin, indicated that increased fucoxanthin might enhance the ability of light absorption. Results of 77 K fluorescence spectra also showed that increased fucoxanthin improved the light capture efficiency of algae (Figure 3B).

Xanthophyll cycle-related NPQ is one of the most important photoprotection mechanisms, and protects plastids from excessive excitation of photosynthetic pigments, which may lead to over-reduction of the electron transport chain and generation of the harmful ROS [37]. There are two different xanthophyll cycles in higher plants and diatom, namely the violaxanthin cycle and the DD-DT cycle, respectively [18]. The DES and (DT+DD)/fucoxanthin of the algae under the treatment of AA were lower than that of CK (Figure 6B), indicating that the DD-DT cycle was affected, which was consistent with the altered NPQ (Figure 6A). It is speculated that it is the changes in the content of DD, DT and fucoxanthin influence the DD-DT cycle, leading to decreased NPQ. LHCXs have been identified as a key component of qE in diatom, and related to NPQ induction, which is similar to LHCSR in higher plants and the green algae [22]. Four LHCX family members have been identified in the genome of *P. tricornutum* [22], and we found that the expression levels of LHCXs in AA-treated algae were all increased, especially the mRNA expression level of LHCX1, comparing with that of CK (Figure 6C), while the protein expression level of LHCX1 was decreased under the treatment of AA (Figure 6D). There are multiple *LHCX* genes in the diatom genomes analyzed so far, which strongly suggests that the expansion of this gene family is a common feature of these algae and may represent an adaptive feature to cope with highly variable environment conditions [22]. As LHCX1 has a crucial role in NPQ regulation and light acclimation [38], the increased mRNA expression level may be to rescue the decreased NPQ. In addition, post-translation modification might be existed as the expression level of mRNA was inconsistent with that of protein.

The xanthophyll cycle is the most important short-term photoprotection mechanism of diatom, which reduces the excitation energy reaching PSII and avoids the photoinactivation of D1 protein [30]. A strong decrease in NPQ induction and lower conversion of DD to DT in AA-treated algae were observed (Figure 6), which may cause severe photoinhibition, as the value of Y(NO) and photoinhibition curve shown (Figure 7). In a word, in AA-treated *P. tricornutum*, increased fucoxanthin content improves the ability of light capture and increases the excitation pressure in the electron transport chain, in the meantime, the decrease in the ability of the DD-DT cycle and the less effective dissipation of excitation energy leads to the occurrence of photoinhibition. In order to adapt the changeable environment of ocean, diatoms own FCPs with capabilities of light harvesting and photoprotection [39]. Pigments Chl *c* and fucoxanthin in FCPs provide a robust energy-quenching system to keep the diatoms from the photodamage when the x-ray crystal structure of a dimeric FCP from *P. tricornutum* was published [39]. In addition, one Dd was found in a position next to the monomer-monomer of FCP [39]. It reveals the DD-DT cycle participates in energy dissipation. In addition, FCP aggregation might cause a conformational change with the formation of a charge transfer state of Chl-fucoxanthin in the oligomeric antenna complexes [29]. Thus, fucoxanthin has a great role in photoprotective function [18]. Previously, it was reported that two different quenching sites under high light illumination were found, quenching site 1 (Q1) and quenching site 2 (Q2) in the pennate *P. tricornutum* and the centric *Cyclotella menghiniana* [40]. Q1 has a Chl *a* fluorescence component with a long wavelength, which can detach and aggregate FCP complexes [18,31]; while the Q2 quenching site distributes in an antenna, which attaches to the photosystem II(PSII) [41]. Interestingly, Q2 is related to the formation of Dt [40]. Moreover, Dt can promote the formation of the Q1 site [18]. Therefore, Q1 quenching site and Q2 quenching site can all promote NPQ. In our study, we found that NPQ was decreased as a consequence of decreased DD-DT cycle when the production of fucoxanthin increased. The increased production of fucoxanthin might cause the conformational change of FCP aggregation. In addition, the decreased DD-DT cycle might change the Q1 and Q2 quenching sites, leading to the decreased NPQ. Besides, our study also investigated the relationship between production of fucoxanthin and the abilities of photosynthesis, photoprotection using exogenous chemical elicitor AA.

Taken together, our results reported an induction of both cell growth and fucoxanthin accumulation by a low concentration of AA. Gene expression analysis proved that such an induction was a result of the transcriptional regulation and also suggested a possible stress triggered by AA treatment. In addition, NPQ was decreased as a consequence of the decreased DD-DT cycle, triggering the occurrence of photoinhibition under the treatment of AA. With a full genome sequence available, further transcriptome analysis should shed light on the discovery of fucoxanthin biosynthesis pathway, and also enable the promotion of fucoxanthin production by genetic engineering.

## 4. Materials and Methods

### 4.1. Algal Strain and Growth Conditions

The *phaeodactylum tricornutum* strain used in this study was maintained on plates with sterilized *f/2* medium (final concentration: 1 × 10^−5^ M FeCl_3_·6H_2_O, 1 × 10^−5^ M Na_2_EDTA·2H_2_O, 4 × 10^−8^ M CuSO_4_·5H_2_O, 3 × 10^−8^ M Na_2_MoO_4_·2H_2_O, 8 × 10^−8^ M ZnSO_4_·7H_2_O, 5 × 10^−8^ M CoCl_2_·6H_2_O, 9 × 10^−7^ M MnCl_2_·4H_2_O) containing 1.0% agar as an axenic culture. Single algal colony was picked and inoculated in sterilized *f/2* mediumin flasks. Cells were grown at 25 °C under a constant irradiation of 62.5 μmol photons m^−2^ s^−1^ with a 12 h/12 h light/dark regime, shaking at 120 rpm. AA was dissolved in ethanol (100 mg/mL) and supplied to the culture to different final concentrations (0, 0.1, 0.5, 2.5, 12.5, 62.5 and 312.5 mg/L, respectively), with the solvent as a control. For each treatment, at least 3 replicates were performed.

### 4.2. Growth Measurement and Fucoxanthin Quantification

The growth of *P. tricornutum* cells was determined by measuring the absorbance of the culture at 680 nm using a spectrophotometer (UV-5200, China). Cell numbers were also counted using a hemocytometer under a microscope.

For fucoxanthin quantification, algal cells were pelleted from an 80 mL culture by centrifugation at 12,000× *g* at 4 °C. Harvested cells were frozen-dried, weighed and then ground into fine powder. One mL anhydrous ethanol was added to each of the algal samples, and the extraction was performed by vigorous vortex for 3 times (30 s each). The extraction was repeated three times. After each extraction, the mixture was centrifuged at 13,000× *g* at 4 °C for 10 min and the supernatants were combined, filtered through a 0.22-µm pore size membrane (Sangon Biotech, Shanghai, China) and then stored at 4 °C in the dark for further analysis.

Carotenoids in the extract were separated by a 2695 high performance liquid chromatography (HPLC) Separations Module (Waters, Milford, MA, USA). Fucoxanthin was detected by monitoring the absorbance at 445 nm using a 486 Tunable Absorbance Detector (Waters). A YMC Carotenoid HPLC Column (250 mm × 4.6 mm with 5-µm particle size) (YMC, Kyoto, Japan) was used and the elution was performed using a gradient of methanol/water ratio from 90/10 to 100/0 over 30 min, followed by a further 20 min with 100% methanol. 

### 4.3. Molecular Manipulation

Total RNA was extracted from algal cells using RNAiso Plus (Takara, Osaka, Japan) and treated by DNase I (Takara, Osaka, Japan) to remove contaminated genomic DNA. Single-stranded cDNA was synthesized from 1 μg of total RNA using the PrimeScript 1st Strand cDNA Synthesis Kit (Takara, Osaka, Japan).

The sequences of genes encoding lycopene β-cyclase (LCYB), PSY, PDS, CRTISO, ZDS, ZEP, and β-ACTIN from the model organism *Chlamydomonas reinhardtii* and *Arabidopsis thaliana* were downloaded from the GenBank and used as queries to blast in GenBank for corresponding homologs in *P. tricornutum* using the tblastX algorithm. Sequence comparison was performed using MEGA 5.0 (Mega Limited, Auckland, New Zealand) [42].

### 4.4. Gene Expression Analysis

Quantitative real-time PCR (qPCR) was performed using Applied Biosystems 7500 Real-Time System (Applied Biosystems, Foster City, CA, USA). All primers used in this study were listed in Appendix A. qPCR amplification program was 40 cycles of 95 °C for 30 s, 53 °C for 15 s, and 72 °C for 20 s. Melting curves and negative controls without cDNA templates were used to detect nonspecific amplification. The expression of *β-ACTIN* was quantified as a reference. Three independent biological samples were adopted with three replicates each. The 2^−^^∆∆CT^ method was used to calculate the relative expression levels [43].

### 4.5. Protein Extraction and Western Blot Analysis

1 × 10^8^ algae cells were collected by centrifugation and frozen in liquid nitrogen immediately. 50 μL lysis buffer (50 mmol/L Tris pH 6.8, 2% SDS) was used to suspend the frozen pellets, and incubated at room temperature for 30 min. Soluble fraction was collected by centrifugation (14,000 rpm) for 30 min at 4 °C. Protein concentration was detected according the guideline of the BCA^TM^ Protein Assay Kit (Sangon Biotech, Shanghai, China). The total protein with same content was transferred to polyvinylidene fluoride (PVDF, Thermo Scientific, Waltham, USA) membrane by wet transfer apparatus (Bio-Rad, California, USA). The antibody against LHCX was obtained from Prof. Wenda Wang (Institute of Botany, the Chinese Academy of Sciences).

### 4.6. Measurement of Chlorophyll Fluorescence

Chlorophyll fluorescence of the algae cells (7.5 mg/mL Chl concentration) were recorded using the MAXI version of the IMAGING-PAM M-Series chlorophyll fluorescence system (Heinz-Walz, Effeltrich, Germany) after darkness adaptation for 20 min. The parameters *F_v_/F_m_* (the maximal quantum yield of PSII), and Y(II) (the effective quantum yield of PSII) were calculated as the following Equations (1) and (2) [44]:*F_v_*/*F_m_* = (*F_m_* − *F_0_*)/*F_m_*(1)
Y(II) = (*F_m_’* − *F_t_*)/*F_m_’*(2)

*F_0_* represents the minimal fluorescence under the dark-adapted state. *F_m_* represents the maximal fluorescence, which was measured by applying the saturating pulse (SP) under the dark-adapted state. *F_m_’* represents the maximum fluorescence signal after the onset of the actinic light (AL). *F_t_* represents the steady-state fluorescence recorded during AL illumination.

Y(I) (the effective quantum yield of PSI), NPQ, and Y(NO) (quantum yield of non-regulated energy of PSII) were measured by DUAL PAM-100 Chl fluorometer (Heinz-Walz, Effeltrich, Germany). The algal strains were qualified to the same Chl concentration (7.5 mg/mL) and adapted in the dark for 20 min before measurement. The parameters were calculated according to the Equations from (3) to (5) [44]:Y(I) = (*P_m_’* − *P*)/*P_m_*(3)
NPQ = (*F_m_* − *F_m_’*)/*F_m_’*(4)
Y(NO) = *F*/*F_m_*(5)
where *P_m_* represents the maximal P700 change, and *P_m_’* represents the maximum P700^+^ signal under AL without far-red (FR) light.

PSI activity was also measured using a Joliot-type LED pump-probe spectrophotometer (JTS-10, Biologic SAS, Claix, France) as described [45] on the same Chl concentration (7.5 mg/mL).

### 4.7. The Detections of PSI Activity, PSII Activity and Chl Fluorescence Emission Spectra

A spectrometer equipped with an integrating sphere unit (JASCO, Tokyo, Japan) was used to detect the absorption spectra between 400–750 nm after the algae were dark-adapted for 20 min. Fluorescence Spectrophotometer (F-7000 FL, Hitachi) was employed to detect the Chl fluorescence emission spectra. The excitation wavelength was set to 540 nm and spectra were recorded at an interval of 0.2 nm between 600–800 nm. The algae on the same Chl concentration (7.5 mg/mL) were analyzed at 77 K.

### 4.8. Photoinhibition Analysis

To analyze the photoinhibition, the MAXI version of the IMAGING-PAM M-Series chlorophyll fluorescence system (Heinz-Walz, Effeltrich, Germany) was employed to record the value of *F_v_/F_m_* at the interval of 10 min under high light treatment of 1000 μmol photons m^−2^ s^−1^ for 60 min. For recovery analysis, the high light-treated algae were then incubated in the darkness, and the value of *F_v_/F_m_* was recorded at the interval of 10 min during following recovery for 60 min.

### 4.9. Statistical Analysis

Principal component analysis (PCA) was performed using SAS 9.0 (SAS Institute Inc., Cary, NC, USA). To determine statistical significance, we employed one-way ANOVA followed by the Student’s *t*-test. Then, normality by D’Agostino-Pearson test and homoscedasticity by Brown-Forsythe test were achieved.

## 5. Conclusions

In this study, we studied the growth, fucoxanthin production, photosynthesis activity and photoprotection of the diatom *P. tricornutum* under the treatment of exogenous arachidonic acid treated as chemical elictors. It demonstrated that exogenous AA induced fucoxanthin production in *P. tricornutum* by promoting carotenoid metabolism at the transcriptional level. In addition, increased fucoxanthin content affected the induction of non-photochemical quenching (NPQ), which may be derived from the changed diadinoxanthin and diatoxanthin cycle, and as a result, the algae grown under the treatment of AA was prone to subject to photoinhibition. This study will provide the mechanism that regulates fucoxanthin biosynthesis and photoprotection in the diatom *P. tricornutum* under the treatment of exogenous arachidonic acid.

## Figures and Tables

**Figure 1 marinedrugs-20-00644-f001:**
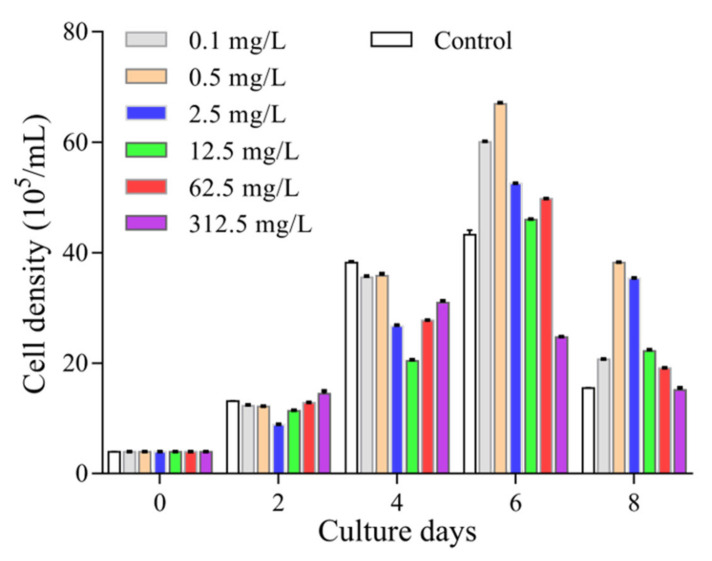
Effect of arachidonic acid (AA) on the growth of *P. tricornutum*. AA was dissolved in ethanol (100 mg/mL) and supplied to the culture as indicated concentrations. Cell numbers were counted using a hemocytometer under a microscope.

**Figure 2 marinedrugs-20-00644-f002:**
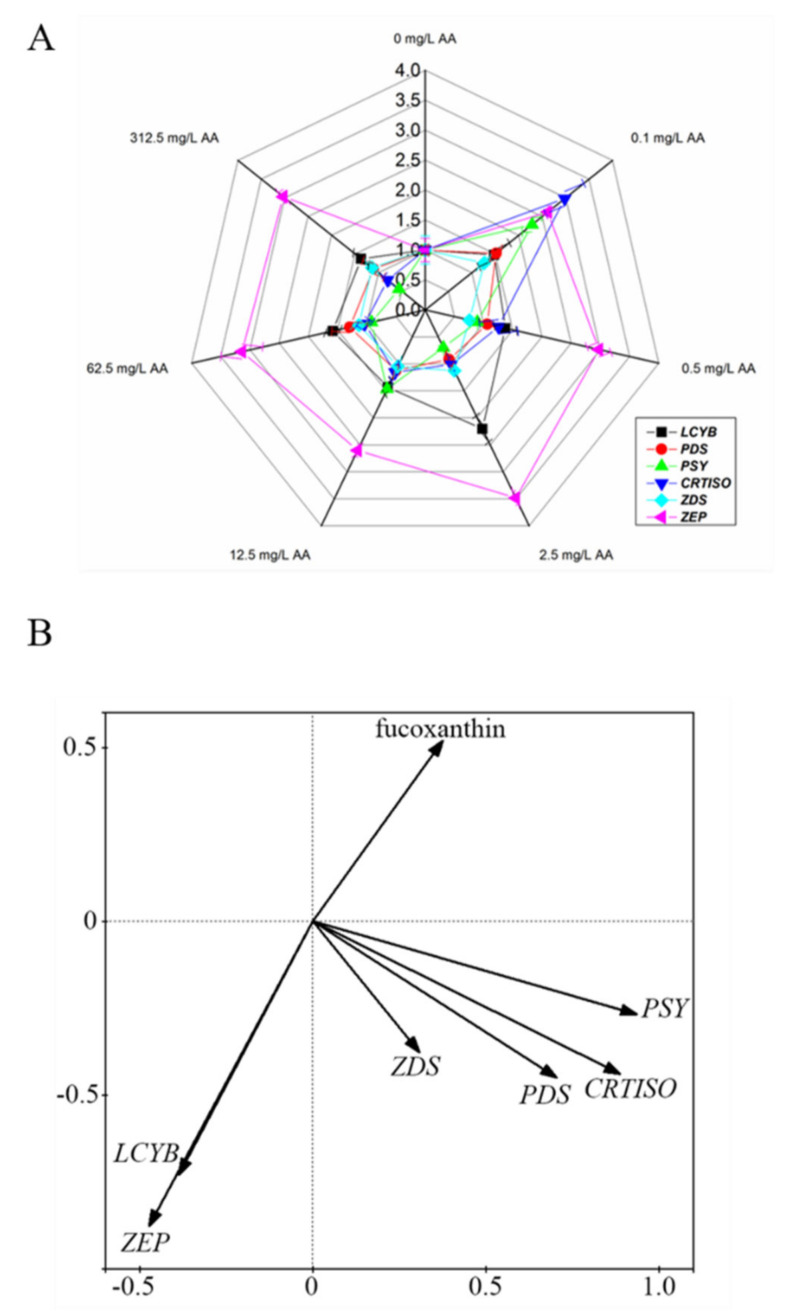
The relationship between the content of fucoxanthin and the transcript abundances of genes for carotenoid metabolism in *P. tricornutum*. (**A**) Transcript abundances of genes for carotenoid metabolism in *P. tricornutum* under the treatment of different concentrations of arachidonic acid (AA) for 6 days. Data represent means ± SD (Student’s *t*-test; *n* = 3). (**B**) Principal component analysis (PCA) of the expression of carotenoid metabolic genes and fucoxanthin production.

**Figure 3 marinedrugs-20-00644-f003:**
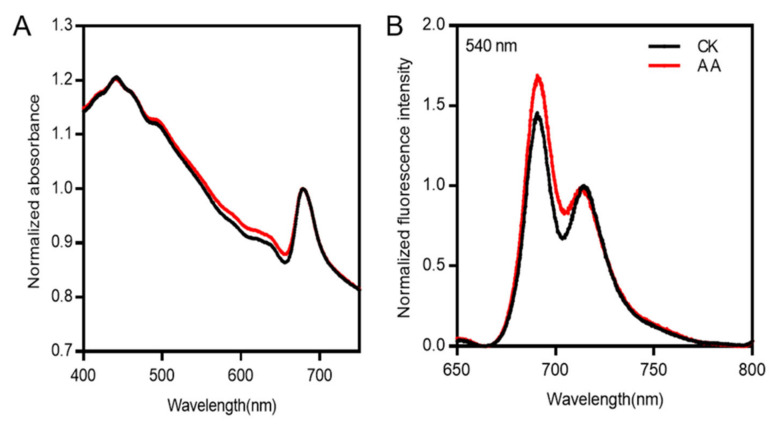
Absorption spectrum in the regions of 400–750 nm (**A**) and 77 K fluorescence emission spectra excited at 540 nm (**B**). (**A**) The black line and the red line represent the spectra of algae with and without arachidonic acid (AA) treatment (0.1 mg/L), respectively, normalized by the peak intensity of Chl *a*. (**B**) The black line and the red line represent the spectra of control check (CK) and AA treatment (0.1 mg/L), respectively. The spectra of CK and AA were normalized at 713 nm.

**Figure 4 marinedrugs-20-00644-f004:**
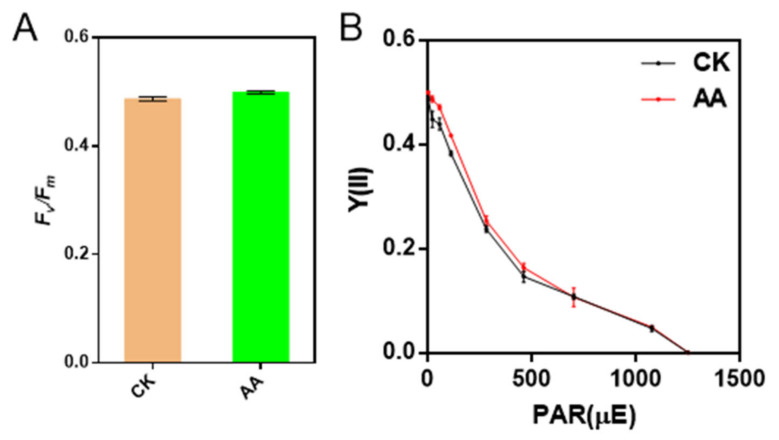
The detection of the activity of photosynthesis II of control check (CK) and arachidonic acid (AA) treatment (0.1 mg/L). (**A**,**B**) represent the photosynthetic parameters of CK and AA, including maximal photochemical efficiency-*F_v_/F_m_* and actual photochemical efficiency-Y(II), respectively. Data represent means ± SD (Student’s *t*-test; *n* = 3).

**Figure 5 marinedrugs-20-00644-f005:**
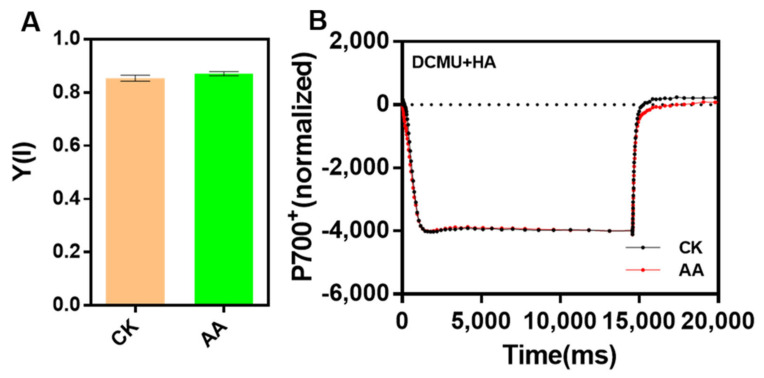
The function of PSI under arachidonic acid (AA) treatment is not impaired. (**A**) The measurement of Y(I) (the effective quantum yield of PSI) activity of control check (CK) and AA treatment (0.1 mg/L) with DUAL-PAM 100 according to the curve of saturation pulse method, which can determine the efficiency of energy conversion around PSI. Data represent means ± SD (Student’s *t*-test; *n* = 3). (**B**) P700^+^ oxidation-reduction kinetics of CK and AA treatment (0.1 mg/L) normalized to total oxidizable P700 in the presence of 3-(3,4-dichloroprenyl)-1-1-dimethylurea (10 μM DCMU) and hydroxylamine (1 mM HA) by using Joliot-type LED pump-probe (JTS-10) spectrophotometer, which can check the PSI activity and block the PSII activity completely.

**Figure 6 marinedrugs-20-00644-f006:**
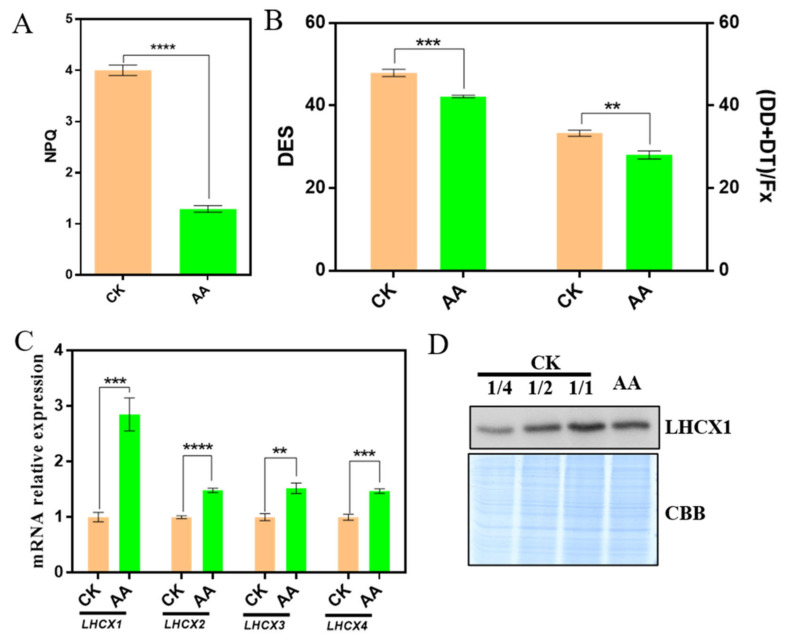
LHCXs may regulate the change of non-photochemical quenching (NPQ) under the treatment of arachidonic acid (AA). (**A**) NPQ assay of control check (CK) and AA treatment (0.1 mg/L) with DUAL-PAM 100. The dark adaption for 30 min before exposure was necessary to measure the *F_m_* and *F_m_’*, the maximum fluorescence emission. The cells were exposed to 1000 μmol photons m^−2^ s ^−1^ high light and a saturation pulse per 20 s. (**B**) High performance liquid chromatography (HPLC) analysis was used to determine the de-epoxidation state (DES) and relative level of the pool of xanthophyll, [diadinoxanthin (DT)+ diatoxanthin (DD)]/fucoxanthin (Fx). (**C**) The mRNA relative expression of four *LHCXs* in CK and AA treatment (0.1 mg/L) with qRT-PCR. *β-ACTIN* was treated as reference gene. (**D**) Accumulation of LHCX1 protein with western blotting. The antibody against the LHCX1 was used to detect the level of LHCX1 protein in both CK and AA treatment (0.1 mg/L). CBB represented the stained gel with Coomassie Brilliant Blue R250 as reference. Data represent means ± SD (Student’s *t*-test; *n* = 3; ** *p* < 0.01; *** *p* < 0.001; **** *p* < 0.0001).

**Figure 7 marinedrugs-20-00644-f007:**
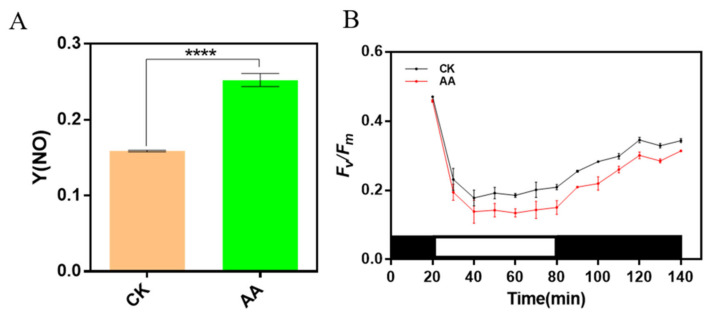
The detection of photoinhibition. (**A**) The detection of Y(NO) (quantum yield of non-regulated energy of PSII) in control check (CK) and arachidonic acid (AA) treatment (0.1 mg/L). Data represent means ± SD (Student’s *t*-test; *n* = 3; **** *p* < 0.0001). (**B**) The detection of *F_v_/F_m_*. The algae were under the dark for 20 min, then the algae were under 1000 μmol photons m^−^^2^ s^−^^1^ high light for 60 min, and the value of *F_v_/F_m_* was recorded at the interval of 10 min. Finally, the algae were under the dark to recover for 60 min, and the value of *F_v_/F_m_* was recorded every 10 min.

**Table 1 marinedrugs-20-00644-t001:** Fucoxanthin production in *P. tricornutum* under different concentrations of arachidonic acid (AA) for 6 days.

AA Concentration (mg/L)	Content of Fucoxanthin [mg/g Dry Weight (DW)]	
	Average	SD
0	0.805 ^a^	±0.190
0.1	1.099 ^a^	±0.205
0.5	0.865 ^a^	±0.033
2.5	0.775 ^a^	±0.156
12.5	0.797 ^a^	±0.275
62.5	0.774 ^a^	±0.095
312.5	0.271 ^b^	±0.003

^a,b^ Different letters represent the significant differences according to ANOVO (*p* < 0.05).

**Table 2 marinedrugs-20-00644-t002:** The highest similarities of carotenoid metabolism genes from *P. tricornutum* with homologs from other organisms.

Gene	Best Hit Organism (Similarity)	Accession Number
*LCYB*	*Fistulifera solaris* (65%)	GAX21707.1
*PSY*	*Fistulifera solaris* (70%)	GAX17065.1
*PDS*	*Chromochloris zofingiensis* (60%)	ABR20877.1
*CRTISO*	*Fistulifera solaris* (70%)	GAX26939.1
*ZDS*	*Fistulifera solaris* (87%)	GAX15703.1
*ZEP*	*Fistulifera solaris* (84%)	GAX11479.1

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
