# Peer review of "Exogenous Arachidonic Acid Affects Fucoxanthin Biosynthesis and Photoprotection in Phaeodactylum tricornutum"

_marinedrugs, 2022, doi:10.3390/md20100644_

Round 1
Reviewer 1 Report
The manuscript is well written, and scientific facts support most conclusions.
However, I would like the author to remove the claim that AA directly impacts the biosynthesis of fucoxanthin. The authors found that the addition of AA significantly changes the growth and metabolism of the diatom Phaeodactylum tricornutum. However, the authors have no direct evidence that AA directly triggers fucoxanthin production. It could be that AA activates the overall metabolism expression of diatom by some general regulator or by the effect of stress, which would increase fucoxanthin production. I would also ask the authors to remove any claims that the paper brings any more evidence in the biosynthetic mechanism of fucoxanthin as an example in the abstract " To understand better the molecular mechanism of fucoxanthin biosynthesis" (no experiments were performed to corroborate those claims).
Author Response
Point 1: However, I would like the author to remove the claim that AA directly impacts the biosynthesis of fucoxanthin. The authors found that the addition of AA significantly changes the growth and metabolism of the diatom Phaeodactylum tricornutum. However, the authors have no direct evidence that AA directly triggers fucoxanthin production. It could be that AA activates the overall metabolism expression of diatom by some general regulator or by the effect of stress, which would increase fucoxanthin production. I would also ask the authors to remove any claims that the paper brings any more evidence in the biosynthetic mechanism of fucoxanthin as an example in the abstract " To understand better the molecular mechanism of fucoxanthin biosynthesis" (no experiments were performed to corroborate those claims).
Response 1: Thank you for your comment. I have modified the section of abstract and improved any claims that no experiments were performed to corroborate.
Reviewer 2 Report
Paper title: Exogenous Arachidonic Acid Affects Fucoxanthin Biosynthesis 2 and Photoprotection in Phaeodactylum tricornutum
This paper studies the influence of exogenous arachidonic acid on the fucoxanthin biosynthesis in a diatom during growth. Overall, the article is well written and organized. It is important though that the suggestions/questions presented below are resolved.
Specific comments:
Keywords: All keywords should be different from title to increase article’s searchability.
Line 100: Why is “P. tricornutum” in several place not written in italics?
Line 117: Make sure that dry weight as “DW” is correctly defined the first time it appears in the text.
Line 121: It is not clear: At which AA concentration is the increase in fucoxanthin production significant? Please revise the paragraph, it is confusing.
Line 124: Add different letters for different averages – interpretation would be easier. If statistics were done as described, why are not different averages indicated in the Table 1?
Line 195: Please increase the size of the y-axis title.
Line 205: Please, avoid starting phrase with “to”. Please rewrite.
Line 204: It is important to note that in this results section there are far too many abbreviations. In this page for example there are easily more than 10. It turns the text difficult to read. Would you consider writing some of these abbreviations in full?
Line 229: Also, in this Figure and all the others make sure that all the abbreviations are written in full, cause Figures and Figure captions must be understood independently from the text.
Line 267: This phrase is very confusing. Please rewrite.
Line 280: There is a space between “Table” and “1”.
Line 305: What agrees? Avoid writing “this” and not specify what.
Line 356: Please specify “f/2” medium.
Lines 365-367: Were controls without P. tricornutum but with AA solution in ethanol tested to assess the UV contribution of AA in ethanol?
Line 417: “were recorded…”, please correct.
Line 441: I suggest renaming the section to “UV spectra” or something similar. The objective of the spectrometric measurement should be title, not the name of the technique. So what was it used for?
Line 452: Why was the interval of 60 min chosen?
Line 455: More information is needed regarding the statistical analysis. Firstly, for one to run a one-way ANOVA and the Student’s t test, data should be validated as parametric. Was the data checked? If so, which tests were used to verify the normality and homoscedasticity of it? Secondly, give more explanations to how PCA was done, which data was considered in each component, which data was used, etc.
Author Response
Point 1: Keywords: All keywords should be different from title to increase article’s searchability.
Response 1: Thank you for your comment. I have changed the keywords.
Point 2: Line 100: Why is “P. tricornutum” in several place not written in italics?
Response 2: Thank you for your comment. I have checked the whole paper and corrected this issue.
Point 3: Line 117: Make sure that dry weight as “DW” is correctly defined the first time it appears in the text.
Response 3: Thank you for your comment. I have defined the dry weight as “DW” for the first time.
Point 4: Line 121: It is not clear: At which AA concentration is the increase in fucoxanthin production significant? Please revise the paragraph, it is confusing.
Response 4: Thank you for your comment. I have revised this paragraph. A low concentration of AA (0.1 mg/L) induced the fucoxanthin production to 1.1 mg/g DW.
Point 5: Line 124: Add different letters for different averages – interpretation would be easier. If statistics were done as described, why are not different averages indicated in the Table 1?
Response 5: Thank you for your comment. I have added different letters for different averages for statistics in the Table 1.
Point 6: Line 195: Please increase the size of the y-axis title.
Response 6: Thank you for your comment. The y-axis of Figure 5(A) title represents the PS(I) activity, which is the ratio of (Pm’-P) and Pm , Where Pm represents the maximal P700 change, and Pm’ represents the maximum P700+ signal under AL without far-red (FR) light. The y-axis of Figure 5(B) represents the P700+ oxidation-reduction kinetics, which is normalized to total oxidizable P700. Thus, the size of the y-axis title cannot be added.
Point 7: Line 205: Please, avoid starting phrase with “to”. Please rewrite.
Response 7: Thank you for your comment. I have modified this issue.
Point 8: Line 204: It is important to note that in this results section there are far too many abbreviations. In this page for example there are easily more than 10. It turns the text difficult to read. Would you consider writing some of these abbreviations in full?
Response 8: Thank you for your comment. I have rewritten some of abbreviations in full to make the text easy to read in the result of 2.6, such as NPQ, DD,DT,ROS and FCP.
Point 9: Line 229: Also, in this Figure and all the others make sure that all the abbreviations are written in full, cause Figures and Figure captions must be understood independently from the text.
Response 9: Thank you for your comment. I have checked all the figures and tables, making sure that all the abbreviations are written in full.
Point 10: Line 267: This phrase is very confusing. Please rewrite.
Response 10: Thank you for your comment. I have rewritten this phrase, making this phrase more clearer.
Point 11: Line 280: There is a space between “Table” and “1”.
Response 11: Thank you for your comment. I have ommited the space between “Table” and “1”.
Point 12: Line 305: What agrees? Avoid writing “this” and not specify what.
Response 12: Thank you for your comment. I have modified this sentence to “Thus, the result of qPCR and PCA analysis suggested that LCYB expression was not correlated with fucoxanthin production.”
Point 13: Line 356: Please specify “f/2” medium.
Response 13: Thank you for your comment. I have specified the final concentions of different components of medium.
Point 14: Lines 365-367: Were controls without P. tricornutum but with AA solution in ethanol tested to assess the UV contribution of AA in ethanol?
Response 14: Thank you for your comment. AA storage solution in ethanol was prepared, and low volume of ethanol was added to the medium. We also checked the values of UV of the media with and without ethanol, there is no different.
Point 15: Line 417: “were recorded…”, please correct.
Response 15: Thank you for your comment. I have corrected this issue.
Point 16: Line 441: I suggest renaming the section to “UV spectra” or something similar. The objective of the spectrometric measurement should be title, not the name of the technique. So what was it used for?
Response 16: Thank you for your comment. I have modified the title of this section, clarifying the objective of the spectrometric measurement.
Point 17: Line 452: Why was the interval of 60 min chosen?
Response 17: Thank you for your comment. The changes of PSII activity (Fv/Fm) during high light treatment (1000 μmol photons m-2 s-1) and subsequent dark recovery were also measured to monitor the photoinhibition process. After dark recovery of 40 min, the values of PSII activity (Fv/Fm) was in stablity for 20 min. Thus, the interval of 60 min was chosen to dark recovery.
Point 18: Line 455: More information is needed regarding the statistical analysis. Firstly, for one to run a one-way ANOVA and the Student’s t test, data should be validated as parametric. Was the data checked? If so, which tests were used to verify the normality and homoscedasticity of it? Secondly, give more explanations to how PCA was done, which data was considered in each component, which data was used, etc.
Response 18: Thank you for your comment. To determine statistical significance, we employed one-way ANOVA followed by the Student’s t-test. Then, normality by D'Agostino-Pearson test and homoscedasticity by by Brown-Forsythe test were achieved.PCA was used to explain the relationship between the mRNA expression level of carotenoid and the content of fucoxanthin. In this study, the mRNA expression of six genes in carotenoid biosynthesis were treated as components to calculate the contribution rate to the content of fucoxanthin using SAS 9.0 (SAS Institute Inc., Cary, NC, USA).
Reviewer 3 Report
Exogenous Arachidonic Acid Affects Fucoxanthin Biosynthesis and Photoprotection in Phaeodactylum tricornutum
By Shuaiqi Zhu et al.
Manuscript ID: marinedrugs-1914805
The research carried out is interesting. The manuscript is very well written and organized. I have a number of comments, with the intention of improving it.
L80
“LHCX”
It is necessary to indicate the meaning of the acronym used.
L100
P. Tricornutum
In italics. Italics should also be used throughout the manuscript for the scientific name of the species (L107, L109, L113 ... and others)
General comment about Figures and Tables
The information presented in the tables and figures must be fully understood without the need to consult the main text of the manuscript. In this sense, it is necessary to specify the acronyms' meaning (both in tables and figures)
L132-134
Scientific names of species should be presented in italics.
L382-385
Related to Fucoxanthin Quantification
Please provide the range of the calibration curve, the number of points and the concentration of the calibration standards used. Also, the regression curve obtained needs to be presented, as well as the limit of detection (LOD) and limit of quantification (LOQ) of the HPLC method.
Author Response
Point 1: L80 “LHCX” It is necessary to indicate the meaning of the acronym used.
Response 1: Thank you for your comment. LHCX proteins are closely related to LHCSR proteins and found in ‘red lineage’ organisms that contain secondary red plastids, such as diatoms. We searched the related papers, the meaning of the acronym of LHCX was not shown except the definition that mentioned above. So we don’t know the meaning of “X”.
Point 2: L100 P. Tricornutum In italics. Italics should also be used throughout the manuscript for the scientific name of the species (L107, L109, L113 ... and others)
Response 2: Thank you for your comment. I have checked the whole paper and corrected this issue.
Point 3: General comment about Figures and Tables
The information presented in the tables and figures must be fully understood without the need to consult the main text of the manuscript. In this sense, it is necessary to specify the acronyms' meaning (both in tables and figures)
Response 3: Thank you for your comment. I have checked all the figures and tables, making sure that all the abbreviations are written in full.
Point 4: L132-134 Scientific names of species should be presented in italics.
Response 4: Thank you for your comment. I have checked the whole paper,making sure that scientific names of species were presented in italics.
Point 5: L382-385 Related to Fucoxanthin Quantification
Please provide the range of the calibration curve, the number of points and the concentration of the calibration standards used. Also, the regression curve obtained needs to be presented, as well as the limit of detection (LOD) and limit of quantification (LOQ) of the HPLC method.
Response 5: Thank you for your comment. Six points were chosen to constructed the calibration curve with the range from 0.002 to 0.2 g/mL. Our statistical analysis resulted in a linear equation of y=0.1074x+0.0138(y represents the peak areas in the spectra, x represents the concentration of fucoxanthin relation coefficient R2=0.9999), which was shown in the method. All points detected were distributed on the calibration curve. While, the limit of detection(LOD) and limit of quantification(LOQ) of the HPLC method were not performed.
Reviewer 4 Report
Dear Authors,
The present study evaluates fucoxanthin biosynthesis and photosynthetic phenotypes in Phaeodactylum tricornutum under the treatment of exogenous arachidonic acid. The research subject is interesting and brings scientific important data in the field, as it deals with a subject that is currently of great interest. Some changes of the manuscript should nevertheless be performed in order to improve its quality. Following specific changes should thus be performed:
Minor changes
Names of all species should be italic. Please check throughout the whole manuscript (e.g. line 100, 107, 109, 113, 124, 127-137 etc.).
Major changes
Introduction: This part should contain information regarding similar studies existing in scientific literature. These information are few. Please insist on purposes of these studies. Moreover, in comparison with these ones, authors should emphasize the novelty and originality of their study. If the presented studies are singular, please mention to emphasize your novelty and originality. If not, please give additional details and state what you bring in novelty. The purpose of the study needs to be rephrased to become clearer and needs to be found in the last paragraph. In fact, this paragraph should not contain a presentation of results, but a clear presentation of your purposes. Please add further information and justifications and modify accordingly. Please offer a rationale for choosing the particular diatom and give more information about it (importance, uses etc.). Make connections between these aspects and offer more details about them in the context of your study.
Discussions: Here you should emphasize novelty and originality of the present study once again, but in a different way than in Introduction. You need to compare your results with the ones obtained by other authors and you need to highlight what you bring in novelty compared to these.
Conclusions: Please add this section. It is necessary, as your Results and Discussions sections are quite long. Please offer perspectives of your study.
All these suggested changes should be performed in order to bring further improvements to the manuscript.
Author Response
Point 1: Minor changes
Names of all species should be italic. Please check throughout the whole manuscript (e.g. line 100, 107, 109, 113, 124, 127-137 etc.).
Response 1: Thank you for your comment. I have checked the whole paper and corrected the issue.
Point 2: Major changes
Introduction: This part should contain information regarding similar studies existing in scientific literature. These information are few. Please insist on purposes of these studies. Moreover, in comparison with these ones, authors should emphasize the novelty and originality of their study. If the presented studies are singular, please mention to emphasize your novelty and originality. If not, please give additional details and state what you bring in novelty. The purpose of the study needs to be rephrased to become clearer and needs to be found in the last paragraph. In fact, this paragraph should not contain a presentation of results, but a clear presentation of your purposes. Please add further information and justifications and modify accordingly. Please offer a rationale for choosing the particular diatom and give more information about it (importance, uses etc.). Make connections between these aspects and offer more details about them in the context of your study.
Discussions: Here you should emphasize novelty and originality of the present study once again, but in a different way than in Introduction. You need to compare your results with the ones obtained by other authors and you need to highlight what you bring in novelty compared to these.
Conclusions: Please add this section. It is necessary, as your Results and Discussions sections are quite long. Please offer perspectives of your study.
All these suggested changes should be performed in order to bring further improvements to the manuscript.
Response 2: Thank you for your comments. In order to emphasize the novelty and originality of my study, the last paragragh in the section of introduction was modified to make my purposes more clearer. In the section of discussions, the first paragragh was the new one to clarify the novelty of my study through comparing other authors’ work. In order to offer perspectives of my study more clearer, the section of conclusions was added after the section of materials and methods.
Round 2
Reviewer 1 Report
Changes have been performed accordingly to the comments.
Author Response
Thank you for your comments.
Reviewer 4 Report
Dear Authors,
The present study evaluates fucoxanthin biosynthesis and photosynthetic phenotypes in Phaeodactylum tricornutum under the treatment of exogenous arachidonic acid. Authors performed some of the suggested changes after the first round of review. However, following specific changes should still be performed:
Major changes
Introduction: I did not find informations regarding purposes of similar studies existing in scientific literature. Therefore, novelty and originality is not clear at all. The last paragraph should not contain a presentation of results. I did not find a rationale for choosing the particular diatom and more information about it.
Discussions: I did not find a comparison of your results with the ones obtained by other authors and neither what you bring in novelty compared to these is emphasized.
Conclusions: This section is poor and does not highlight the results of your study and neither its perspectives.
All these suggested changes should be performed in order to bring further improvements to the manuscript.
Author Response
Response to Reviewer 4 Comments
Point 1: Major changes
Introduction: I did not find informations regarding purposes of similar studies existing in scientific literature. Therefore, novelty and originality is not clear at all. The last paragraph should not contain a presentation of results. I did not find a rationale for choosing the particular diatom and more information about it.
Discussions: I did not find a comparison of your results with the ones obtained by other authors and neither what you bring in novelty compared to these is emphasized.
Conclusions: This section is poor and does not highlight the results of your study and neither its perspectives.
All these suggested changes should be performed in order to bring further improvements to the manuscript.
Response 1: Thank you for your comment.
Introduction:
According to your comment, I have added the reason of choosing the P. tricornutum to perform the experiment. In addition, I have deleted the presentation of results in the last paragraph. Finally, I introduce the purpose of this study. These contents were all described in the last paragraph.
Discussions:
I have further compared the expression levels of genes in the carotenoid biosynthesis with other studies when the content of fucoxanthin was increased. This content has been added in the second paragraph.
I have also added the extra paragraph(paragraph 8 in discussions) to discuss the function of fucoxanthin in the aspect of photoprotection. Previously, it was reported that FCP aggregation might cause a conformational change with the formation of a charge transfer state of Chl-fucoxanthin in the oligomeric antenna complexes. Thus, fucoxanthin has a great role in photoprotective function. In addtion, two qunching sites, Q1 and Q2, have been found to promote the NPQ in the pennate P. tricornutum and the centric Cyclotella menghiniana. While, the issue that the production of fucoxanthin induces NPQ by inducing DD-DT cycle has not been mentioned. Hence, this is the novelty of this study.
Conclusions:
In this section, I have improved the results of my study by adding the purpose of the study. In addtion, the perspective is more cleared in the last sentence of this section.
Round 3
Reviewer 4 Report
Dear Authors,
The present study evaluates fucoxanthin biosynthesis and photosynthetic phenotypes in Phaeodactylum tricornutum under the treatment of exogenous arachidonic acid. Authors performed most of the suggested changes after the second round of review. However, following specific changes should still be performed:
Major changes
Introduction: Novelty and originality is still not very clear.
All these suggested changes should be performed in order to bring further improvements to the manuscript.
